# Liver Ultrasound Elastography in Non-Alcoholic Fatty Liver Disease: A State-of-the-Art Summary

**DOI:** 10.3390/diagnostics13071236

**Published:** 2023-03-24

**Authors:** Rosanna Villani, Pierluigi Lupo, Moris Sangineto, Antonino Davide Romano, Gaetano Serviddio

**Affiliations:** 1Liver Unit, C.U.R.E. (University Centre for Liver Disease Research and Treatment), Department of Medical and Surgical Sciences, University of Foggia, 71122 Foggia, Italy; 2Department of Radiology, University of Foggia, 71122 Foggia, Italy

**Keywords:** liver, ultrasound, elastography, NAFLD, steatosis, steatohepatitis

## Abstract

Non-alcoholic fatty liver disease (NAFLD) is a chronic disease which is currently the most common hepatic disorder affecting up to 38% of the general population with differences according to age, country, ethnicity and sex. Both genetic and acquired risk factors such as a high-calorie diet or high intake of saturated fats have been associated with obesity, diabetes and, finally, NAFLD. A liver biopsy has always been considered essential for the diagnosis of NAFLD; however, due to several limitations such as the potential occurrence of major complications, sampling variability and the poor repeatability in clinical practice, it is considered an imperfect option for the evaluation of liver fibrosis over time. For these reasons, a non-invasive assessment by serum biomarkers and the quantification of liver stiffness is becoming the new frontier in the management of patients with NAFLD and liver fibrosis. We present a state-of-the-art summary addressing the methods for the non-invasive evaluation of liver fibrosis in NAFLD patients, particularly the ultrasound-based techniques (transient elastography, ARFI techniques and strain elastography) and their optimal cut-off values for the staging of liver fibrosis.

## 1. Introduction

Non-alcoholic fatty liver disease (NAFLD) is the most common chronic liver disease worldwide [1].

The overall prevalence of NAFLD in the general population has been increased over time from 25% before 2005 to 38% in 2016 and later, and it is significantly higher (up to 80%) in the diabetic population [2,3,4].

NAFLD is histologically defined as the abnormal accumulation of fat (>5%) in the hepatocytes in the absence of secondary causes of fatty liver disease, such as significant alcohol consumption or viral infection. The severity of liver steatosis is graded as S0 when histological involvement is minimal (<5% of hepatocytes), S1 or mild (5–33% of hepatocytes), S2 or moderate (34–66%) and, finally, S3 or severe when >66% of liver cells show the intrahepatic accumulation of lipids [5].

NAFLD is a term which encompasses the simple and benign deposition of fat in hepatocytes to more progressive and aggressive NASH (non-alcoholic steatohepatitis) characterized by hepatitis and fibrosis up to cirrhosis and hepatocellular carcinoma (HCC) [6].

A recent meta-analysis found that the overall mean prevalence of significant fibrosis (F2–F4), advanced fibrosis (F3–F4) and cirrhosis is 45.0%, 24.0% and 9.4%, respectively, in NAFLD patients; these percentages are only a little lower than those reported in patients with viral hepatitis (56.6%, 33.5% and 18.4%, respectively, in HBV patients; 57%, 35% and 13% in HCV patients) [7].

Despite the rising interest and gains in our understanding of NAFLD/NASH pathogenesis over the last two decades, there has been some dissatisfaction with the terminology “non-alcoholic” because it overstresses the role of alcohol and plays down the role of metabolic risk factors in the development of the disease. Therefore, a name change from NAFLD to metabolic-associated fatty liver disease (MAFLD) has been proposed, even if it is still considered unsatisfactory because it gives more relevance to the metabolic risk factors without addressing the physiopathology of this liver disease. This is why some authors have suggested to use the term MAFLD with caution because changing the name without understanding its implications could have a negative impact on the field [8].

The diagnosis of NAFLD is based on the evidence of hepatic steatosis either by imaging or histology, whereas liver biopsy is always required for the diagnosis of NASH [9].

NASH is distinguished from isolated and benign hepatic steatosis (non-alcoholic fatty liver, NAFL) by the presence of hepatocellular injury, lobular inflammation and hepatocellular ballooning with or without liver fibrosis [10].

In 25% of patients, NAFLD progresses to NASH, which associates with an increased risk of cirrhosis and hepatocellular carcinoma [11]. In patients with NASH, the stage of fibrosis is the most important determinant of liver-related progression and mortality [12], and this is why the assessment of liver fibrosis is a key step for the optimal management of patients with NAFLD.

Liver biopsy has always been considered the gold standard for staging liver fibrosis; however, it is an invasive procedure associated with rare but potentially major complications [13], and sampling errors may easily occur due to the limited parenchymal tissue (1/50,000th of the total liver mass) [14].

Moreover, the error rate in staging fibrosis occurs in 20% of cases, cirrhosis is not correctly diagnosed in 20% of samples and the interobserver variability is about 10% [15,16,17].

All of these issues make liver biopsy an imperfect option for evaluating liver fibrosis [18].

Therefore, several methods have been studied for the non-invasive assessment of liver fibrosis, including serum biomarkers and the quantification of liver stiffness (LS).

Among the different techniques for the non-invasive assessment of liver stiffness in chronic liver disease, ultrasound elastography has given the most important contribution to the non-invasive assessment of liver fibrosis in NAFLD patients and several manuscripts have been published in the last years. Therefore, we performed a review including the most recent literature addressing this hot topic in hepatology.

## 2. Ultrasound-Based Techniques for the Assessment of Liver Fibrosis: Principles and Systems

Non-invasive methods for the assessment of liver fibrosis are one of the fields that has most rapidly evolved in the last years due to the limitations of liver biopsy and the need for several re-evaluations of liver fibrosis over time.

Blood markers, as a non-invasive method for the staging of liver fibrosis, have low accuracy in discriminating among intermediate stages of fibrosis because several extra-hepatic conditions may interfere or significantly influence them [14,19].

Additionally, the liver is a large parenchymal organ readily accessible to US scanning and elastography measurements [20] and, moreover, these methods can also be easily repeated without the risk of complications or need for post-procedure hospitalization [14].

This is why the interest in non-invasive ultrasound-based techniques has increased progressively as shown by the current medical literature.

These techniques, which differ from the physical approach used, include three major types of non-invasive methods: transient elastography (TE), which uses a mechanical external push; acoustic radiation force impulse (ARFI) techniques, which use an acoustic internal push; and, finally, the strain elastography (SE) based on the tissue deformation caused by pressing the body surface or by using the internal physiologic movements (breathing and heartbeat). The ARFI techniques can be divided into point shear wave elastography (p-SWE) and 2D-shear wave elastography (2D SWE) techniques [21]. TE and ARFI techniques measure the speed of shear waves in tissues [20].

Transient elastography (TE) is a 1D technique performed with the FibroScan (Echosens, Paris, France). Fibroscan has three types of probes (S, M, XL) which work with different ultrasound frequencies. The S probe, for the pediatric population, uses a frequency of 5.0 MHz for measurements at a depth of between 1.5 and 5.0 cm from the body surface. The M probe uses an ultrasound frequency of 3.5 MHz (depth from 2.5 to 6.5 cm from the skin) and, finally, the XL probe is able to reach a depth of 3.5–7.5 cm thanks to an ultrasound frequency of 2.5 MHz [21].

The ARFI techniques include the point shear wave elastography (p-SWE) (Figure 1) and 2D-shear wave elastography (2-D SWE) (Figure 2) techniques. Both of these measure the speed of shear waves in tissues generated by a push pulse of a focused ultrasound beam. The shear wave speed expressed in m/s is then converted into kilopascals, the unit of Young’s modulus E (3rv2, where r is the tissue density and v is the speed of the shear wave), assuming that the tissue is elastic, that the tissue density is a constant value (1000 kg/m) and that the elastic modulus is not influenced by the frequency and direction of the applied force [21,22].

The stimulation is performed at a definite depth and generates shear waves that propagate perpendicularly to the axis of the push pulse. The shear wave velocity is measured in a definite region of interest (ROI) chosen by the operator during the B-mode examination. The speed of shear waves is assessed by measuring the time required to reach a specific point starting from an opposite point (lateral side) of the region of interest. The higher the shear wave velocity, the higher the tissue stiffness [22].

The 2D-SWE method is based on the combination of acoustic radiation force by focused ultrasonic beams and very high frame ultrasound imaging. The location of the regions of interest (ROIs) and their size can be chosen by the operator before obtaining the stiffness value [21]. A real-time 2D color map (elastogram) is superimposed on the B-mode image and most vendors provide a confidence map to confirm the quality of the recorded values. Three or five measurements are required if the US system has a map confirming that the area chosen for the assessment of liver fibrosis has high-quality shear waves.

After recording the stiffness values, the average, the standard deviation, the minimum and maximum stiffness values can be obtained. The standard deviation within the ROI reports the variability of the pixel measurements within the ROI and it is not a measure of the quality of the measurement [21].

The strain elastography (SE) technique (Figure 3) uses frame-to-frame differences (tissue deformation) with stress caused after pressing the body surface with the probe or using the internal physiologic motion.

Using an intercostal space as a window, the strain image is obtained by the pressure exerted on the liver by cardiac movement. The patient should stop breathing in mid-inspiration/expiration during the acquisition. The main limitations are the somatic features because it can be difficult to obtain an optimal histogram in obese patients or in patients with shortness of breath [20].

## 3. Liver Ultrasound Elastography in NAFLD Patients: Transient Elastography

Several studies are available on the performance of TE for the assessment of liver fibrosis in NAFLD patients. All studies including at least 50 patients and published during the last 10 years (between January 2012 and December 2023) addressing the topic of the use of TE in NAFLD patients are reported in Table 1. Three of them are meta-analyses published in 2017, 2021 and 2022, respectively.

The first meta-analysis was published by Xiao et al. in 2017, who addressed the interesting hot topic of the identification of the best method for diagnosing liver fibrosis in NAFLD patients and, particularly, the authors compared the performance of aspartate aminotransferase to the platelets ratio index (APRI), fibrosis-4 index (FIB-4), BARD score, NAFLD fibrosis score (NFS), FibroScan, shear wave elastography (SWE) and magnetic resonance elastography (MRE) [7].

Data were analyzed both for the M and XL probe (N = 13,046). Generally, transient elastography did not show a good accuracy in staging liver fibrosis in NAFLD patients because of a wide range in sensitivity and specificity and overlapping values for different stages of liver fibrosis (see Table 1).

The best range for the identification of advanced fibrosis was 7.6–9 kPa with 83% to 89% sensitivity and 77% to 78% specificity [7]. For the detection of liver fibrosis, the M probe showed a sensitivity and specificity of 91.9% and 55.5%, respectively, at a cut-off value of 5.8 kPa, whereas 6.65–7 kPa had a sensitivity and specificity of 80.1% and 68.3%, respectively [7].

Selvaraj et al. performed a new large meta-analysis in 2021 including 82 studies (14,609 patients) [24]. In total, 53 of them addressed the diagnostic performances of TE in the detection of the fibrosis stage. A cut-off value of 8.7 KPa was the best threshold for the identification of significant fibrosis (F3–F4). However, studies included in the meta-analysis used both M and XL probes, and a specific cut-off for different probes was not reported [24].

For the XL probe, the cut-off proposed for the detection of advanced fibrosis ranged from 5.7 to 9.3 kPa. These data were obtained from three studies including 579 patients, with 75% sensitivity and 74% specificity [21].

The last meta-analysis available was published by Cao et al. in 2022 who found the following cut-off values for the identification of significant fibrosis (F2), advanced fibrosis (F3) and cirrhosis (F4): 7.8 KPa, 9.9 KPa and 13.2 KPa, respectively. The proposed cut-offs showed a wide range with overlapping values (see Table 1).

This study included 10,537 patients from 61 studies and found different liver stiffness measurement (LSM) values among different regions; for diagnosing stages ≥ F2 and F4, the mean cut-off values of European and American patients were 0.96 and 2.03 kPa higher than Asiatic ones [25].

It is noteworthy that most available studies included in our review showed heterogenous data because populations were studied by using the M or XL probe and final cut-offs were a combination of their results.

It has been reported that the use of the XL probe is associated with lower cut-off values (−2 KPa) in comparison to the M probe, and this suggests that different cut-off values should be defined. The most recent meta-analysis published by Cao et al. in 2022 showed that the cut-off value for the detection of significant fibrosis by using the M probe was 7.8 KPa, which is similar to the threshold always used for the staging of liver fibrosis in patients with chronic HCV infection [26], whereas, in line with the lower values obtained by using the XL probes, and reported for the first time in patients with chronic viral hepatitis, the use of 6.9 KPa as a threshold value can be useful for the diagnosis of significant fibrosis (F2 stage) in obese patients. The same authors proposed 12.2 and 13.2 KPa as cut-off values for the diagnosis in NAFLD patients.

We identified 24 studies analyzing only data from populations studied by M probe. The best sensitivity of 100% for the identification of liver cirrhosis was recorded by Attia et al. and Chang et al. and Chan et al. who proposed a cut-off value of 15 KPa. Populations of their studies came from Malaysia, Singapore and Germany, suggesting that this threshold may be used for Asian and Caucasian patients. The highest sensitivity was also recorded by Lee H.W. et al. in a Korean population; however, their best cut-off was lower (11 KPa). Only two studies studied the best cut-off for the identification of liver cirrhosis in obese patients who underwent a liver stiffness measurement by using the XL probe.

Wong et al. in 2019 used the same cut-off value for the identification of liver cirrhosis (15 KPa) and found that sensitivity was only 49% even if specificity was high (93%) (see Table 1) [27].

On the other hand, Attia et al. found that 11.9 KPa was the best cut-off for the identification of patients (sensitivity 100%) with a good specificity (93%) [28].

The impact of liver steatosis on the cut-off values irrespective of liver fibrosis is a challenging issue. Petta et al. in 2015 published the results of a multi-center study including NAFLD patients who underwent liver biopsy [29]. In total, 6.9 kPa and 8.4 kPa were the best cut-off values for F2 and F3 liver fibrosis. Interestingly, the authors showed that the presence of obesity and severe steatosis was associated with higher stiffness values, suggesting a potential overestimation of fibrosis due to the liver steatosis [29].

This observation was not confirmed by more recent studies that did not find a strict association between the severity of liver steatosis and liver stiffness measurements [30], and confirmed by Wong et al. who showed that BMI but not severity of steatosis increased liver stiffness and that the same LSM cut-offs could be used without further adjustment when M and XL probes were used according to the BMI subgroups [27].

Particularly, these authors found that liver stiffness assessed with the XL probe was 2.3 KPa lower than values obtained with the M probe; however, patients with a BMI ≥ 30 kg/m^2^ had similar liver stiffness values regardless of the probe used, suggesting that when M and XL probes are used in subgroups of patients by BMI (<30 vs. ≥30 kg/m^2^), identical stiffness measurements are obtained with similar accuracy and diagnostic performance [27]. However, this limitation could be overcome by using the XL probe, which generally gives cut-off values 1.5–2 KPa lower than those found by using the M probe. These results were suggested by several observations reported in the literature showing that values obtained with M and XL probes have a comparable diagnostic accuracy, even if the stiffness values obtained by using XL probes are lower [31,32,33]. However, even if the probe does not affect the liver stiffness measurement [30], the use of the XL probe can improve the reliability of TE on the condition that specific cut-off values should be considered when XL probes are used [33].

The cut-off values proposed by different authors for the identification of significant fibrosis (≥F2) were highly variable and showed a wide range of sensitivity.

In 2017, Lee M.S. et al. enrolled 94 patients with biopsy-proved NAFLD and found that 7.4 and 8.0 KPa as cut-off values for the identification of liver fibrosis F2 and  ≥F3 by using the M probe had the following sensitivities: 62.5%, and 82.6%, respectively; specificities were 91.7% and 84.9%, respectively [34].

On the other hand, by using the M probe and a cut-off value of 8.2 KPa for the stage F2, Cardoso et al. obtained a higher sensitivity compared to that reported by Lee M.S. et al. (93.3% versus 62.5%), although they used a cut-off which was only slightly different.

Similarly, Eddowes et al. in 2019 suggested the following cut-off values for fibrosis stage ≥F2 and ≥F3: 8.2 kPa and 9.7 kPa, respectively [30].

In 2022, Argalia et al. assessed LSM in a small population (*n* = 50) with NAFLD and found that the median liver stiffness measurements for fibrosis stages F1, F2 and F3 were 5.5 (4.4–7.3) kPa, 7.7 (6.1–9.1) kPa and 9.9 (8.8–13.8) kPa, respectively [35].

In 2016, Imajo et al. enrolled 142 patients studied by using the M probe and with biopsy-proven NAFLD and found that 11 KPa and 11.4 KPa were the cut-off values for the correct staging of intermediate liver fibrosis [36].

Essentially, different experiences worldwide have shown that the intermediate stage of liver fibrosis (F2 and F3) may have similar cut-off values making its diagnosis difficult and showing that transient elastography is probably not a better method for the non-invasive diagnosis of intermediate stage liver fibrosis in patients with NAFLD.

**Table 1 diagnostics-13-01236-t001:** Studies addressing the use of TE and cut-off values for staging liver fibrosis in NAFLD patients published in the last 10 years.

Study	Country	Study Design	Aim of the Study	Patients (*n*)	Probe	Cut-Off Values (KPa)	SE (%)	SP (%)	AUROC (95% CI or DS)
Cao et al., 2022 [25]	China	MA	Accuracy of LSM for assessing fibrosis	10,537	M	≥F1 6.63 (5.3–7.5)			
						≥F2 7.82 (5–11)			
						≥F3 9.91 (7.1–13.6)			
						F4 13.26 (9.5–17.5)			
					XL	≥F2 6.95 (5–8.9)			
						≥F3 9.24 (7.2–11.5)			
						F4 12.26 (7.9–17.5)			
Roccarina et al., 2022 [37]	Italy UK	Prospective	pSWE vs. TE for diagnosis of fibrosis stage	671	M/XL	≥F1 6.6	85	69	0.79 (0.60–0.91)
						≥F2 8.5	83	70	0.85 (0.78–0.91)
						≥F3 10.6	76	81	0.85 (0.79–0.91)
						F4 12.5	89	83	0.91 (0.83–0.96)
Argalia et al., 2022 [35]	Italy	Prospective	Comparison TE/pSWE	50	M/XL	≥F1 4.23	82.7	57.1	0.72 (0.57–0.83)
						≥F2 4.63	73.9	62.9	0.73 (0.59–0.90)
						≥F3 7.39	87.5	88.1	0.91 (0.79–0.97)
						F4 14.	100	100	1.00 (0.93–1.00)
Mikolasevic et al., 2021 [38]	Croatia	Prospective	Diagnostic accuracy of the CAP and TE	179	M/XL	≥F1 6.7	74.8	91.6	0.830
						≥F2 8.2	85.2	91.2	-
						≥F3 10	97.6	92.6	0.98
						F4 13.4	94.7	99.3	0.98
Selvalaj et al., 2021 [24]	UK	MA	Diagnostic accuracy of TE, pSWE, 2D-SWE and MR	1064	M/XL	≥F1 5.3–8.2	78	72	0.82 (0.78–0.85)
						≥F2 3.8–10.2	80	73	0.83 (0.80–0.87)
						≥F3 6.8–12.9	80	77	0.85 (0.83–0.87)
						F4 6.9–19.4	76	88	0.89 (0.84–0.93)
Trowell et al., 2021 [39]	US	Prospective	Assessment of steatosis staging by CAP scores	92	M/XL	≥F3 11.9	85	69	0.85 (0.77–0.92)
Yang et al., 2021 [40]	China	Retrospective	Diagnostic accuracy of TE in patients with abnormal glucose metabolism and impact of metabolic indicators on the LSM value	91	M	≥F1 6.3	71.1	75	0.79 (0.69–0.87)
						≥F2 7.6	68	68.3	0.76 (0.66–0.85)
						≥F3 8.3	80	76.1	0.84 (0.74–0.91)
						F4 13.8	80	94.2	0.90 (0.88–0.95)
Taibbi et al., 2021 [41]	Italy	Prospective	pSWE vs. TE for LSM	56	M/XL	≥F3 7.9	63	63.2	0.72 (0.57–0.87)
						F4 8.5	77.8	78.6	0.80 (0.65–0.95)
Sharpton et al., 2021 [42]	US	Prospective	Diagnostic accuracy of 2D-SWE vs. TE	114	M/XL	F2 ≥ 6.8	94.6	62.3	0.86 (0.80–0.93)
						F3 ≥ 8.7	95	80.9	0.91 (0.82–0.99)
						F4 10.6	100	80	0.96 (0.91–1.00)
Shima et al., 2020 [43]	Japan	Retrospective	Diagnostic accuracy of combined biomarker measurements and TE for predicting fibrosis stage	278	M	F1 ≥ 7.2	81.3	78.5	0.855
						≥F3 9.9	83.2	86.2	0.891
Oeda et al., 2020 [33]	Japan	Prospective	Accuracy of probes M and XL	104	M/XL	≥F2 7	93	63	0.780
						≥F3 10.8			
						F4 16.8			
Shi et al., 2020 [44]	Japan	Prospective	estimation of the optimal cut-off values of LSM in non-obese patients	158	M	≥F1 7.5	71	88.9	0.87 (0.81–0.92)
						≥F2 8.5	84.3	85.5	0.89 (0.83–0.93)
						≥F3 10.8	83.3	83.7	0.89 (0.83–0.94)
						F4 13.1	88.5	82.6	0.90 (0.85–0.94)
Forsgren et al., 2020 [45]	Sweden	Prospective	Comparison of multimodal MR, serum algorithms and TE	90	M/XL	≥F3 10.15	86	84	0.84 (0.71–0.97)
Leong et al., 2020 [46]	Malaysia	Prospective	Comparing pSWE and TE for diagnosis of fibrosis stage	100	M/XL	≥F1 7.68	83.3	81.3	0.89 (0.81–0.97)
						≥F2 9.13	87.8	66.1	0.83 (0.74–0.91)
						≥F3 9.28	90.9	64.2	0.83 (0.75–0.91)
						F4 13.45	100	76	0.89 (0.80–0.99)
Jafarov et al., 2020 [47]	Turkey	Retrospective	Diagnostic utility of fibrosis-4 score LSM for the assessment of advanced liver fibrosis	139	M/XL	≥F2 8.95	76	59	0.72 (0.63–0.80)
						≥F3 22	84	78	0.86 (0.79–0.92)
Furlan et al., 2020 [48]	US	Prospective	Comparison of 2D-SWE, TE and MR elastography for the diagnosis of fibrosis	62	M/XL	≥F2 8.8	51.2	94.4	0.77 (0.65–0.89)
						≥F3 6.7	86.4	70.3	0.86 (0.77–0.95)
Cardoso et al., 2020 [49]	Brazil	Prospective	Performance of CAP and TE comparing XL with M probes	81	M	≥F2 8.2	93.3	63.6	0.78 (0.68–0.86)
					XL	≥F2 8.2	73.3	77.2	0.75 (0.64–0.84)
Tovo et al., 2019 [50]	Brazil	Prospective	Validation of the performance of LSM, APRI, FIB4 and NAFLD score in the evaluation of liver fibrosis	104	M/XL	≥F3 7.9	95	58.3	0.87 (0.78–0.97)
						≥F3 8.7	90	64.3	
						≥F3 9.6	85	69	
Staufer et al., 2019 [51]	Austria	Prospective	Comparison of LSM with ELF test, FibroMeter^V2G^, FibroMeter^V3G^, NFS and FIB-4 in prediction of liver fibrosis	186	M/XL	≥F2 8.2	83	68	0.87 (0.80–0.94)
						≥F3 9.7	92	77	
						≥F3 11	90	80	
Hanafy et al., 2019 [52]	Egypt	Prospective	Evaluation of a non-invasive model in the prediction of cardiovascular morbidity and histological severity	272	M/XL	≥F3 9.75	97.8	98	0.88 (0.94–0.97)
Chang et al., 2019 [53]	Singapore	Prospective	Optimal liver stiffness measurement values for the diagnosis of significant fibrosis and cirrhosis	51	M/XL	≥F2 11	94.4	76.7	0.907
						F4 15	100	81	0.950
Eddowes et al., 2019 [30]	UK	Prospective	Accuracy of TE in assessing fibrosis	373	M/XL	≥F2 8.2	71	70	0.77 (0.72–0.82)
						≥F3 9.7	71	75	0.80 (0.75–0.84)
						F4 13.6	85	79	0.89 (0.84–0.93)
Siddiqui et al., 2019 [54]	US	Prospective	Diagnostic accuracy of TE in detection of NAFLD	393	M/XL	≥F1 8.6	53	87	0.74 (0.68–0.79)
						≥F2 8.6	66	80	0.79 (0.74–0.83)
						≥F3 8.6	80	74	0.83 (0.79–0.87)
						F4 13.1	89	86	0.93 (0.90–0.97)
Lee JI et al., 2019 [55]	Korea	Retrospective	TE in prediction of liver fibrosis	184	M	≥F2 8.95	72.5	65.4	0.730
Wong et al., 2019 [27]	China	Prospective	Unified interpretation of liver stiffness measurement by M and XL probes	496	M	≥F2 5	97.4	35.1	0.86 (0.83–0.90)
						≥F3 10	72.7	89	0.86 (0.82–0.89)
						F4 15	46.9	95.5	0.85 (0.80–0.90)
					XL	≥F2 5	91.8	25	0.81 (0.77–0.85)
						≥F3 10	56.8	82.5	0.84 (0.80–0.88)
						F4 15	48.6	93	0.89 (0.85–0.92)
Boursier et al., 2019 [56]	France	Prospective	Combination of non-invasive tests for the diagnosis of advanced fibrosis	938	M/XL	≥F3 7.9	91.1	59.8	0.840 (±0.013)
Lee et al., 2017 [34]	Korea	Prospective	Comparison among TE, supersonic SWE and ARFI	94	M	≥F2 7.4	62.5	91.7	0.76 (0.64–0.87)
						≥F3 8	82.6	84.9	0.87 (0.77–0.96)
						F4 10.8	91.7	81.2	0.88 (0.74–0.93)
Petta et al., 2017 [57]	Italy	Prospective	Combination of non-invasive tools for the evaluation of liver fibrosis	761	M	≥F3 9.6	74	81	0.863
Park et al., 2017 [58]	US	Prospective	MR elastography vs. TE in detection of fibrosis	94	M/XL	≥F1 6.1	66.7	65.1	0.67 (0.56–0.78)
						≥F2 6.9	79.3	84.6	0.86 (0.77–0.95)
						≥F3 7.3	77.8	77.6	0.80 (0.67–0.93)
						F4 6.9	62.5	66.3	0.69 (0.45–0.94)
Petta et al., 2017 [59]	Italy	Prospective	Prediction of liver fibrosis by TE	324	M	≥F2 8.5	74.3	73.7	0.808
						≥F3 10.1			
Chan et al., 2017 [60]	Malaysia	Prospective	CAP using the FibroScan XL probe for quantification of hepatic steatosis	57	M	≥F1 7.1	79.4	80	0.88 (0.78–0.94)
						≥F2 10.7	84.6	89.4	0.95 (0.87–0.98)
						≥F3 13.6	87.5	97.2	0.97 (0.90–0.99)
						F4 15.1	100	96.1	0.97 (0.90–1.00)
					XL	≥F1 5.9	85.3	75.6	0.87 (0.78–0.94)
						≥F2 8.9	44.1	93.3	0.90 (0.81–0.95)
						≥F3 11.5	87.5	97.2	0.95 (0.87–0.98)
						F4 12.4	100	94.7	0.98 (0.91–1.00)
Seki et al., 2017 [61]	Japan	Retrospective	Assessment of liver fibrosis by TE	171	M	≥F1 7.2	78.5	78.3	0.85 (0.78–0.91)
						≥F3 10	89.5	87.6	0.91 (0.83–0.97)
Loong et al., 2017 [62]	China	Prospective	Accuracy and utility of FM TE for fibrosis staging	215	M	≥F2 9	65.2	87.7	0.851 (±0.029)
						≥F3 9.6	83.7	86.6	0.940 (±0.016)
Xiao et al., 2017 [7]	China	MA	Comparison of laboratory tests, ultrasound or MR elastography for the detection of liver fibrosis	13,046	M	≥F2			
						5.8	91.7	57.4	
						6.65–7.10	74.1	68.6	
						7.25–11	65.7	84.5	
						≥F3			
						6.95–7.25	69.2		
						7.6–8	88.9	66.3 77.2	
						8.7–9	83.3	78 89.9	
						9.6–11.4	80.1		
						F4		77.7 86.3 88.8 90.8	
						7.9–8.4	96.5		
						10.3–11.3	87.7		
						11.5–11.95	77.5		
						13.4–22.3	78.2		
					XL	≥F2	75.8	64.8	
						4.8–8.2			
						≥F3	75.3	74	
						5.7–9.3			
						F4	87.8	82	
						7.2–16			
Tapper et al., 2016 [63]	US	Prospective	Performance of TE	164	M	≥F3 9.9	95	77	0.93 (0.86–0.96)
Attia et al., 2016 [28]	Germany	Prospective	LSM using ARFI elastography in overweight and obese patients	87	M	≥F2 7	85	80	0.88 (0.77–0.95)
						≥F3 11.8	79	94	0.88 (0.77–0.95)
						F4 15	100	93	0.97 (0.89–0.99
					XL	≥F2 6.7	87	76	0.79 (0.59–0.92)
						≥F3 9.3	91	80	0.91 (0.73–0.99)
						F4 11.7	100	83	0.92 (0.75–0.99)
Ergelen et al., 2016 [64]	Turkey	Prospective	Comparison of Doppler ultrasound and TE in the diagnosis of significant fibrosis	63	M/XL	≥F2 9.8	90	91	0.95
Lee HW et al., 2016 [65]	Korea	Prospective	Identification of NASH using TE	183	M	≥F1 6.7	66.4	84.9	0.85 (0.80–0.91)
						≥F2 8	82.6	84.7	0.89(0.83–0.95)
						≥F3 9	96.4	85.8	0.97 (0.95–0.99)
						F4 11	100	89.8	0.97 (0.95–0.99)
Cassinotto et al., 2016 [66]	France	Prospective	Comparison of SWE, TE and ARFI vs. liver biopsy for the assessment of LSM	291	M	≥F2 6.2	90	90	0.82 (0.76–0.87)
						≥F3 8.2	90	90	0.86 (0.80–0.90)
						F4 9.5	92	90	0.87 (0.79–0.92)
Cassinotto et al., 2016 [66]	France	Prospective	2D-SWE vs. TE vs. ARFI	291	M	F2 ≥ 6.2 KPa *	90	45	0.82 (0.76–0.87)
						F3 ≥ 8.2 KPa	90	61	0.86 (0.80–0.90)
						F4 9.5 KPa	92	62	0.87 (0.79–0.92)
Imajo et al., 2016 [36]	Japan	Retrospective	TE vs. MRE to assess liver fibrosis	142	M	≥F1 7	61.7	100	0.78 (0.70–0.87)
						≥F2 11	65.2	88.7	0.82 (0.74–0.89)
						≥F3 11.4	85.7	83.8	0.88 (0.79–0.97)
						≥F4 14	100	75.9	0.92 (0.86–0.98)
Ergelen et al., 2015 [67]	Turkey	Prospective	Addition of serum biomarkers to TE and improvement of diagnostic accuracy in patients with biopsy-proven NAFLD	87	M/XL	≥F2 9.6	68	90	0.87 (0.78–0.97)
						≥F3 9.9	86	77	0.91 (0.82–0.99)
Petta et al., 2015 [29]	Italy	Retrospective	Combination of LSM and NAFLD fibrosis score for improving non-invasive diagnostic accuracy	179	M	Cohort 1 ≥F3 9.3	85.3	81.4	0.86 (0.79–0.92)
						Cohort 2 ≥F3 9.3	68	86.4	0.85 (0.77–0.92)
Pathik et al., 2015 [68]	India	Prospective	TE vs. simple non-invasive screening tools in predicting fibrosis in high-risk non-alcoholic fatty liver disease patients	110	M	≥F3 12	90	80	0.91
Kumar et al., 2013 [69]	India	Prospective	Performance of LSM in patients with different stages of NAFLD	120	M	≥F1 6.1	78	68	0.82 (0.75–0.89)
						≥F2 7	77	78	0.85 (0.78–0.92)
						≥F3 9	85	88	0.94 (0.89–0.98)
						F4 11.8	90	88	0.96 (0.92–1.00)
Mahadeva et al., 2013 [70]	Malaysia	Prospective	Factors associated with discordance between liver histology and TE	131	M	≥F2 6.85	58.8	69.2	0.67 (0.57–0.77)
						≥F3 7.1	70.4	66.6	0.77 (0.66–0.87)
						F4 11.3	87.5	89.3	0.95 (0.91–0.99)

* cut-off for predefined sensitivity >90%. TE: transient elastography; LSM: liver stiffness measurement.

## 4. Liver Ultrasound Elastography in NAFLD Patients: Point Shear Wave Elastography (pSWE)

Most studies dealing with the role of pSWE in the management of patients with liver disease included patients with chronic viral hepatitis; however, only few studies have addressed the utility of point shear wave elastography in the assessment of liver fibrosis in patients with NAFLD. Table 2 shows all of the studies published in the last 10 years.

Before this period of time, only two studies addressed this topic in a very small cohort of patients. In 2010, Osaki et al. addressed the usefulness of pSWE in the management of non-alcoholic steatohepatitis. Only twenty-six patients were included in the study, of which twenty-three had NASH, whereas three patients were the controls [71]. The authors suggested that a cut-off value of 1.47 m/s gave a significant contribution for distinguishing stages 3 and 4 from stages 0 and 1 with excellent sensitivity and specificity (100% and 75%, respectively). In 2011, Friedrich-Rust et al. published the first meta-analysis addressing the performance of pSWE for the staging of liver fibrosis and included a small group of NAFLD patients (*n* = 77) from four studies [72]. The overall diagnostic accuracy was optimal for the identification of cirrhosis (0.94 (0.81, 1.00)) in comparison with the diagnostic accuracy of ≥F2 or ≥F3 (0.86 (0.75, 0.96) and 0.86 (0.58, 1.00), respectively).

After this publication, more studies have been published addressing this topic.

As shown in Table 2, cut-off values for the identification of significant fibrosis are highly variable; however, a cut-off value of 14.2 KPa reported by Argalia et al. [35] showed both a sensitivity and specificity of 100%. It is noteworthy that more than one region may be used for the study of liver stiffness. For this purpose, Attia et al. confirmed that both segments 6 and 8 had similar cut-off values, sensibility and specificity [28]. Most parts of the cut-off values obtained in the study realized in the last 10 years included Caucasian populations. Therefore, more studies are needed to confirm that similar thresholds may be used in non-Caucasian people.

Two meta-analyses have been published in 2015 and 2018 by Liu et al. [73] and Jiang et al. [74], respectively; both of them addressed the reliability of pSWE in staging liver fibrosis in NAFLD patients and found that sensibility and specificity were high enough to consider pSWE a good method for the identification of patients with significant fibrosis and cirrhosis.

**Table 2 diagnostics-13-01236-t002:** Studies published in the last 10 years addressing the use of point shear wave elastography in NAFLD patients.

Study	Country	Study Design	Aim of the Study	Patients (*n*)	Cut-Off Values (m/s or KPa)	SE (%)	SP (%)	AUROC (95% CI)
Argalia et al., 2022 [35]	Italy	Prospective	Comparison of pSWE vs. TE	50	≥F1 4.23 KPa	82.7	57.1	0.717 (0.572–0.835)
					≥F2 4.63 KPa	73.9	62.9	0.733 (0.589–0.848)
					≥F3 7.39 KPa	87.5	88.1	0.908 (0.792–0.971)
					≥F4 14.20 KPa	100	100	1.000 (0.929–1.000)
Bauer et al., 2022 [75]	Austria	Prospective	pSWE for fibrosis screening in Patients with NAFLD	332	F2 ≥ 1.47 m/s	80	95	0.940 (0.910–0.969)
					F3 ≥ 1.52 m/s	88	89	0.949 (0.919–0.979)
					F4 1.86 m/s	87	94	0.949 (0.910–0.989)
Roccarina et al., 2022 [37]	Italy UK	Prospective	pSWE vs. TE for diagnosis of fibrosis stage	671	F1 ≥ 6 KPa	79	81	0.84 (0.72–0.93)
					F2 ≥ 8 KPa	78	81	0.83 (0.78–0.90)
					F3 ≥ 9 KPa	79	78	0.86 (0.82–0.93)
					F4 11.9 KPa	92	85	0.95 (0.92–0.99)
Selvalaj et al., 2021 [24]	UK	MA	Diagnostic accuracy of TE, pSWE, 2D-SWE and MR	276	≥F1 1.11–1.81	64	76	0.77 (0.55–0.92)
					≥F2 1.18–1.81	69	85	0.86 (0.78–0.90)
					≥F3 1.34–4.34	80	86	0.89 (0.83–0.95)
					F4 1.36–2.56	76	88	0.90 (0.82–0.95)
Taibbi et al., 2021 [41]	Italy	Prospective	pSWE vs. TE for liver stiffness quantification	56	F3 ≥ 8.4 KPa	74	73.7	0.787 (0.646–0.927)
					F4 9.1 KPa	72.2	78.5	0.797 (0.659–0.935)
Leong et al., 2020 [46]	Malaysia	Prospective	pSWE vs. TE for diagnosis of fibrosis stage	100	F1 ≥ 6.22 KPa	81.3	66.7	0.79 (0.65–0.92)
					F2 ≥ 6.98 KPa	78.1	61.4	0.74 (0.62–0.85)
					F3 ≥ 7.3 KPa	74.1	63.3	0.71 (0.59–0.83)
					F4 11.52 KPa	66.7	93.2	0.72 (0.31–1.00)
Jiang et al., 2018 [74]	China	MA	pSWE for staging hepatic fibrosis	982	-	70	84	0.86 (0.83–0.89)
						89	88	0.94 (0.91–0.95)
						89	91	0.95 (0.93–0.97)
Lee et al., 2017 [34]	Korea	Prospective	Comparison among TE, SWE and ARFI	94	≥F2 1.35 m/s	46.2	93.2	0.65 (0.54–0.75)
					≥F3 1.43 m/s	70	93.7	0.87 (0.77–0.96)
					F4 1.50 m/s	75	90.7	0.92 (0.84–0.99)
Attia et al., 2016 [28]	Germany	Prospective	ARFI in overweight and obese patients	97	Segment 6			
					F2 ≥ 1.17 m/s	86	87	0.90 (0.83–0.97)
					F3 ≥ 1.42 m/s	97	97	0.99 (0.96–1)
					F4 1.89 m/s	90	95	0.98 (0.96–1)
					Segment 8			
					F2 ≥ 1.18 m/s	78	88	0.86 (0.79–0.94)
					F3 ≥ 1.47 m/s	94	97	0.96 (0.89–1)
					F4 1.89 m/s	86	94	0.93 (0.83–1)
Cui et al., 2016 [76]	US	Prospective	MRE versus ARFI for diagnosing fibrosis in patients with biopsy-proven NAFLD	114	F2 ≥ 1.29 m/s	82	78	0.848 (0.776–0.921)
					F3 ≥ 1.34 m/s	95	74	0.896 (0.824–0.968)
					F4 2.48 m/s	78	93	0.862 (0.721–1.000)
Cassinotto et al., 2016 [66]	France	Prospective	2D-SWE vs. TE vs. ARFI	291	≥F2 0.95 m/s	90	36	0.77 (0.70–0.83)
					≥F3 1.15 m/s	90	63	0.84 (0.78–0.89)
					F4 1.30 m/s	90	67	0.84 (0.78–0.89)
Liu et al., 2015 [73]	China	MA	ARFI for the non-invasive evaluation of hepatic fibrosis	723	-	80.3	85.2	0.898
Cassinotto et al., 2013 [77]	France	Prospective	ARFI vs. LSM and FibroTest	321	≥F2 1.38 m/s	71	78	0.77 (0.72–0.82)
					≥F3 1.57 m/s	75	80	0.82 (0.76–0.86)
					F4 1.61 m/s	82	74	0.84 (0.78–0.88)
Fierbinteanu Brati- cevici et al., 2013 [78]	Romania	Prospective	ARFI for non-invasive evaluation of liver fibrosis	64	≥F1 1.105 m/s	76.7	71.4	0.867 (0.782–0.953)
					≥F2 1.165 m/s	84.8	90.3	0.944 (0.891–0.997)
					≥F3 1.48 m/s	86.4	95.2	0.982 (0.956–1.000)
					F4 1.63 m/s	91.7	92.3	0.984 (0.958–1)

## 5. Liver Ultrasound Elastography in NAFLD Patients: 2D-Shear Wave Elastography

Together with MRE, 2D-SWE has been currently considered the method with the highest accuracy for staging liver fibrosis in NAFLD patients [7].

The 2D-SWE has shown a sensitivity and specificity of 90% and 93%, respectively, for the identification of advanced liver fibrosis [7].

However, only a few studies are available in the literature addressing the clinical utility of 2D-SWE for the assessment of liver stiffness in NAFLD patients. We found only 10 studies which are shown in Table 3.

The latest guidelines recommended a potential role for 2D-SWE to rule out advanced fibrosis and for the selection of patients who deserve further assessment [21].

This recommendation was based on the availability of three studies which showed a good performance in patients with advanced fibrosis.

Some additional studies have recently been published (Table 1) and, therefore, they deserve attention in the field. The largest cohort (*n* = 577 patients) was studied by Cassinotto et al. in 2021 who observed that the performances of 2D-SWE, as a first step, were good (accuracy = 82.3%, sensitivity = 88.3%, specificity = 80.9%, NPV = 87.5%, PPV = 76.4% for ≥F3; PPV = 94.2% for ≥F2).

The authors found that, using the same cut-off values for the 2D-SWE and TE for advanced liver fibrosis, the accuracy of this method was good, and the inclusion of 2D-SWE in a three-step strategy (FIB4 +TE+2D-SWE) strongly decreased the need for liver biopsy to <5% of patients who require the invasive approach for the correct classification of liver fibrosis.

**Table 3 diagnostics-13-01236-t003:** Studies published in the last 10 years addressing the use of 2D-shear wave elastography in NAFLD patients.

Study	Country	Study Design	Aim of the Study	Patients (*n*)	Values (KPa or m/s)	SE (%)	SP (%)	AUROC (95% CI)
Zhang et al., 2022 [79]	US	Prospective	Diagnostic performance of 2D-SWE vs. MR elastography	100	F1 ≥ 1.27 m/s	91.2	11.6	0.65 (0.54–0.76)
					F2 ≥ 1.49 m/s	90.5	43	0.81 (0.72–0.89)
					F3 ≥ 1.46 m/s	93.8	39.3	0.81 (0.71–0.91)
					F4 1.59 m/s	100 *	61.7	0.94 (0.89–1.00)
Cassinotto et al., 2021 [80]	France	Prospective	TE vs. 2D-SWE in a multi-step strategy to detect fibrosis	577	F3 ≥ 9.4	88.3	90.9	0.88 (0.84–0.91)
Podrug et al., 2021 [81]	Croatia and Romania	Prospective	Diagnostic performance of 2D-SWE	232	F2 ≥ 7.9	78.7	92.1	0.91 (0.850.94)
					F3 ≥ 10	66.6	91.6	0.92 (0.860.95)
					F4 11.4	80.9	93.4	0.95 (0.910.98)
Lee et al., 2021 [82]	Korea	Prospective	Accuracy of 2D-SWE	102	F1 ≥ 6.3 KPa	63	88	0.87 (0.79–0.93)
					F2 ≥ 7.6 KPa	89	77	0.87 (0.79–0.93)
					F3 ≥ 9 KPa	100	85	0.95 (0.89–0.99)
Sharpton et al., 2021 [42]	US	Prospective	Diagnostic accuracy of 2D-SWE vs. TE	114	F2 ≥ 7.7 KPa	75.7	85.7	0.84 (0.76–0.92)
					F3 ≥ 7.7 KPa	90	77.7	0.88 (0.81–0.96)
					F4 9.3 KPa	88.9	84.8	0.93 (0.86–0.99)
Selvalaj et al., 2021 [24]	UK	MA	Diagnostic accuracy of TE, pSWE, 2D-SWE and MR	488	≥F28.3–11.6 KPa	71	67	0.75 (0.58–0.87)
					≥F3 9.3–13. 1 KPa	72	72	0.72 (0.60–0.84)
					F4 14.4–15.7 KPa	78	84	0.88 (0.81–0.91)
Furlan et al., 2020 [48]	US	Prospective	2D-SWE vs. TE vs. and MR elastography for the diagnosis of fibrosis	62	F2 ≥ 5.7	87.5	70.6	0.80 (0.67–0.92)
					F3 ≥ 8.1	71.4	94.4	0.89 (0.80–0.98)
Herrman et al., 2018 [83]	France	MA	Assessment of liver fibrosis by 2D-SWE	91	F2 ≥ 7.1	93.8	52	0.855 0.917
					F3 ≥ 9.2	93.1	80.9	
					F4 13	75.3	87.8	
Lee et al., 2017 [34]	Korea	Prospective	Comparison among TE, SWE and ARFI	94	≥F2 8.3 KPa	87	55.3	0.75 (0.64–0.85)
					≥F3 10.7 KPa	90	61.2	0.80 (0.69–0.89)
					F4 15.1 KPa	90	78	0.90 (0.81–0.96)
Cassinotto et al., 2016 [66]	France	Prospective	2D-SWE vs. TE vs. ARFI	291	F2 ≥ 6.3 KPa *	90	50	0.86 (0.79–0.90)
					F3 ≥ 8.3 KPa	91	71	0.89 (0.83–0.92)
					F4 10.5 KPa	90	72	0.88 (0.82–0.92)

* cut-off for predefined sensitivity >90%.

## 6. Liver Ultrasound Elastography in NAFLD Patients: Strain Elastography

The role of strain elastography in the staging of liver fibrosis in NAFLD patients is very limited. In 2001, Ogino et al. studied 107 patients and assessed the diagnostic performance of the strain elastography in staging liver fibrosis in comparison with SWE in biopsy-proven NAFLD.

The diagnostic performance of the strain elastography measured by the area under the curve was 0.75 for F2, 0.80 for F3 and 0.85 for F4, whereas the AUROC for SWE was 0.88 for F2, 0.87 for F3 and 0.92 for F4. Therefore, the results showed that strain elastography was inferior to SWE in terms of the ability to classify liver fibrosis in patients with NAFLD [84].

On the other hand, Ochi et al. in 2012 assessed the liver stiffness in 181 patients with real-time tissue elastography in patients with non-alcoholic fatty liver diseases and compared the results with the histological stage. The cut-off values were 2.47 for F1, 2.67 for F2, 3.02 for F3 and 3.36 for F4 with a diagnostic accuracy between 82.6% and 96.0% in all stages [85].

The authors concluded that strain elastography reliably identifies the early stage of fibrosis in NAFLD patients and that, therefore, it is a useful tool for the management of NAFLD patients.

In 2015, Kobayashi et al. published a meta-analysis to study the overall accuracy of strain elastography in the staging of liver fibrosis in patients affected by different liver diseases [86]. The analysis included 15 studies and 1626 patients and showed that, compared with transient elastography and ARFI imaging, the accuracy of strain elastography was similar for evaluating significant liver fibrosis, but less accurate for the identification of patients with cirrhosis [86]. General results were confirmed in a subgroup analysis including only patients with NAFLD.

Finally, general experience in the literature has shown that strain elastography is a tool which has not been playing a key role in the clinical management of patients with NAFLD. However, recently, some authors have proposed a new concept of “combinational elastography” based on the assumption that the combination of strain and shear wave imaging may increase the relevance of each single ultrasound-based elastography and, therefore, improve their accuracy in the correct classification of liver fibrosis [87].

## 7. Conclusions

Ultrasound elastography has become the most important non-invasive tool for the assessment of liver fibrosis in patients with NAFLD. It includes different techniques with different diagnostic performances. Strain elastography seems to have the lowest diagnostic accuracy when it is used alone. Transient elastography and 2D-shear wave elastography have shown good accuracy in diagnosing significant fibrosis; however, their sensibility and specificity are not optimal for detecting low-grade fibrosis yet. Future studies are needed to explain the role of the operator experience on the accuracy of liver ultrasound elastography in detecting intermediate stage liver fibrosis and the impact of the severity of liver steatosis and/or somatic features (obesity or overweight) on the diagnostic performances of the different ultrasound elastography techniques.

## Figures and Tables

**Figure 1 diagnostics-13-01236-f001:**
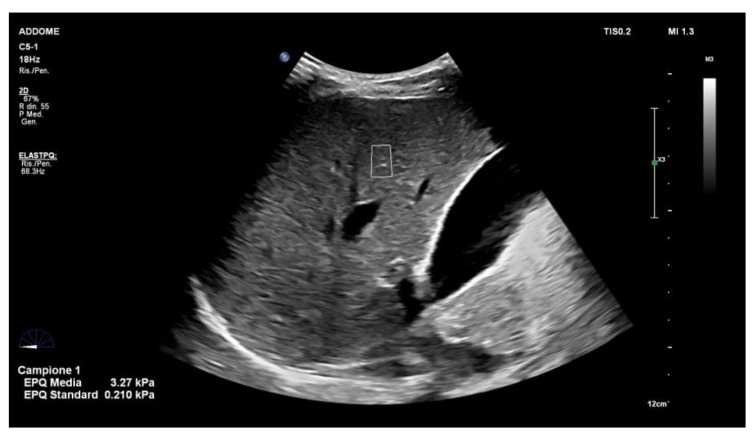
Point shear wave elastography (ElastPQ; Philips Medical System, Best, the Netherlands). The shear wave velocity is measured in the region of interest (ROI) marked by the white line, which can be moved on the screen. Measurements may be expressed as m/s or KPa. Ten measurements should be obtained from 10 independent images, in the same location. The final result should be expressed as the median together with the IQR/M. IQR/M should be ≤30% of the 10 measurements expressed as kilopascals and ≤15% for measurements expressed as meters per second. Measurement should be taken at least 15–20 mm below liver capsule [23].

**Figure 2 diagnostics-13-01236-f002:**
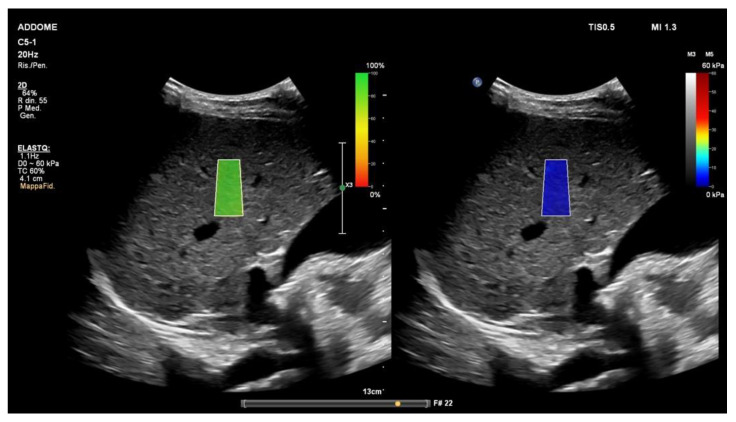
The 2D-shear wave elastography (ElastQ; Philips Medical System, Best, The Netherlands). The color-coded confidence map (**left**) reflects the quality of shear waves. The real-time 2D quantitative color map, namely elastogram (**right**), is superimposed on the B-mode. Even if ten stiffness values obtained from 10 independent images in the same location are generally required for liver ultrasound elastography, when a quality assessment parameter (confidence map) is used, five measurements may be appropriate. The ROI should be positioned at least 15–20 mm below the liver capsule [23].

**Figure 3 diagnostics-13-01236-f003:**
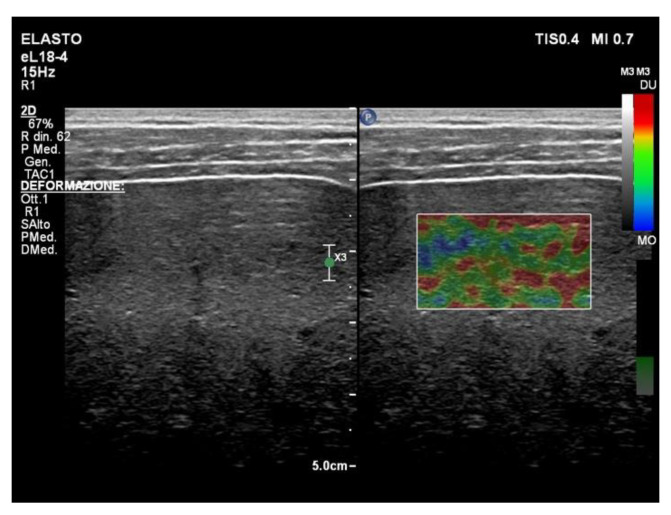
Strain elastography (Philips Medical System, Best, The Netherlands). The screen displays two images: conventional B-mode ultrasound image (**left**) and color-coded elastogram superimposed on B-mode ultrasound image (**right**). The region of interest marked by the white line should avoid large vascular structures.

## Data Availability

Not applicable.

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
