# Peer review of "Liver Ultrasound Elastography in Non-Alcoholic Fatty Liver Disease: A State-of-the-Art Summary"

_diagnostics, 2023, doi:10.3390/diagnostics13071236_

Round 1
Reviewer 1 Report
Review Report
· In this article, the authors aimed to present the state-of-the-art summary addressing the methods for the non-invasive evaluation of liver fibrosis in NAFLD patients, particularly the ultrasound-based techniques (transient elastography, ARFI techniques and strain elastography) and their optimal cut-off values for the staging of liver fibrosis.
· The review is very interesting. The authors have worked hard to present their opinion.
· But I have some comments:
1. Lack of novelty: similar topics were published before and discussed the same points.
2. The aim of the article should be written at the end of the introduction section.
3. Please avoid bullets in writing. Instead, write in paragraphs.
4. Several sentences were without references.
5. Please write a conclusion for this review.
6. Supporting images are required.
Author Response
We are very grateful to the reviewer for the positive comments.
We have revised the manuscript according to his/her suggestions.
Point 1: Abstract: (line 10): The exposed prevalence of NAFLD is much too high, indeed in selected groups of patients such as the diabetic population or in certain geographical areas it can reach up to 50%, but this should not be generalized and the percentage mentioned in the introduction should be exposed, namely 30% according to the current epidemiological data.
Reply: we thank the reviewer for the important comment. We have changed the sentence as suggested.
Point 2: Introduction (Lines 25-34): there are too many paragraphs involving the same topic, namely NAFLD
Reply: Done. We have deleted the extra sentences.
Point 3: Line 252. The exposed studies would be indicated to be in a descending manner depending on the year of publication, being much easier for readers to guide themselves
Reply: We thank the reviewer for the comments. We have changed the text according to his/her suggestion
Point 4: Regarding the NAFLD prevalence I suggest that using Riazi K et al meta-analysis. Lancet Gastroenterol Hepatol. 2022 Sep;7(9):851-861, this being the newest meta-analysis regarding the prevalence of the disease
Reply: We thank the reviewer for the pertinent comment. We have added the reference and concordantly changed the introduction
Reviewer 2 Report
1. The authors reviewed the most recent and relevant articles regarding liver ultrasound elastography in nonalcoholic fatty liver disease. The quality of this manuscript is good, and the conclusions appropriately summarize the data that this study provided. The paper is well organized, briefly addressing the ultrasonographic methods of NAFLD staging, implicitly, using the current literature.
Some revisions are necessary:
MINOR:
- Abstract: (line 10): The exposed prevalence of NAFLD is much too high, indeed in selected groups of patients such as the diabetic population or in certain geographical areas it can reach up to 50%, but this should not be generalized and the percentage mentioned in the introduction should be exposed, namely 30% according to the current epidemiological data.
- Introduction (Lines 25-34): there are too many paragraphs involving the same topic, namely NAFLD
- Line 252. The exposed studies would be indicated to be in a descending manner depending on the year of publication, being much easier for readers to guide themselves
- Regarding the NAFLD prevalence I suggest that using Riazi K et al meta-analysis. Lancet Gastroenterol Hepatol. 2022 Sep;7(9):851-861, this being the newest meta-analysis regarding the prevalence of the disease
English language polishing is needed.
Author Response
Point 1: The review is very interesting. The authors have worked hard to present their opinion.
Reply: we are very grateful for the positive comment
Point 2: Lack of novelty: similar topics were published before and discussed the same points.
Reply: We thank the reviewer for the comment. The topic we addressed is a hot topic in the field of hepatology. For this reason, several manuscript have been published in the last years and this is why we aimed to take stock of the situation. As hepatologists who operate every day using liver ultrasound elastography, we feel the need to have the most updated review in our hands.
Point 3:The aim of the article should be written at the end of the introduction section.
Reply: we thank the reviewer for the comment. We have changed the manuscript as suggested.
Point 4: Please avoid bullets in writing. Instead, write in paragraphs.
Reply: we are grateful for the pertinent suggestion. We have changed the paragraph as suggested
Point 5: Several sentences were without references.
Reply: We thank the reviewer for the careful revision. We have done a check of all sentences and added additional references, as suggested.
Point 6: Please write a conclusion for this review
Reply: Done
Point 7: Supporting images are required
Reply: we thank the reviewer for the pertinent comment. We have added supporting images as suggested.
Round 2
Reviewer 1 Report
The authors have performed a good job and responded to all reviewers comments